# Experimental Prototype of Electromagnetic Emissions for Biotechnological Research: Monitoring Cocoa Bean Fermentation Parameters

**DOI:** 10.3390/foods12132539

**Published:** 2023-06-29

**Authors:** Tania María Guzmán-Armenteros, Jenny Ruales, José Villacís-Chiriboga, Luis Santiago Guerra

**Affiliations:** 1Department of Food Science and Biotechnology, Escuela Politécnica Nacional, Quito P.O. Box 17-01-2759, Ecuador; tania.guzman@epn.edu.ec (T.M.G.-A.); jose.villacis@epn.edu.ec (J.V.-C.); 2Carrera de Medicina, Facultad de Ciencias Médicas, Universidad Central del Ecuador, Quito P.O. Box 17-12-759, Ecuador; lsguerrap@uce.edu.ec

**Keywords:** monitoring, Helmholtz coil, magnetic field, Hall effect sensor, Arduino nano, validation

## Abstract

A Helmholtz-type electromagnetic emission device, which uses an oscillating magnetic field (OMF), with potential applications in biotechnological research, was built and validated. The coils were connected to an alternating current (AC) generator to generate a 0.5 to 110 mT field at their center. OMF measurements were performed with a Hall effect sensor with a digital signal connection (Arduino nano) and data output to a PC using LabVIEW v2017SP1 software. The fermentation process of the cocoa bean variety CCN 51, exposed to four levels of OMF density for 60 min (0, 5, 40, and 80 mT/60 min), was analyzed. Different variables of the grain fermentation process were evaluated over six days. The ANOVA test probed the device’s linearity, accuracy, precision, repeatability, reliability, and robustness. Moreover, CCN 51 cocoa beans’ EMF-exposure effect was evaluated under different OMF densities for 60 min. The results show the validity of the equipment under working conditions and the impact of EMF (electromagnetic fields) on the yield, deformation, and pH of cocoa beans. Thus, we concluded that the operation of the prototype is valid for use in biotechnological studies.

## 1. Introduction

The effects of EMFs (electromagnetic fields) on biological systems have been the subject of extensive research, including their impact on fermentative processes [1,2]. It is known that many biological systems are affected by the presence of a magnetic field, and this process can be approached from a double perspective: firstly through an analysis of the direct action of the EMF study [2] and secondly in the surrounding environment [3].

Recent research indicates that electromagnetic fields can stimulate or inhibit different biological systems [1,2,3,4,5,6,7,8,9,10,11,12,13,14,15,16,17,18], such as the growth of plants [13], fruits [14], and microorganisms [15]. Similarly, they affect cells in the human [16] and animal [17] body by stimulating or inhibiting different cellular processes [18]. Several of these cellular properties have been used to improve health [4], obtain metabolites of industrial interest [5], and remove contaminants from soil [6], water [7], and air [8]. In this sense, electromagnetic fields offer a sustainable and non-invasive solution with different practical applications such as medical therapy [9], production processes [10], and environmental protection [11], maintaining natural integrity [3].

Therefore, three fundamental criteria must be considered when conducting an EMF investigation. The first is to unify evidence-based research criteria, following a single and coherent trend. The second is to ensure experimental validity to obtain reproducible results while maintaining the uniformity of electromagnetic field measurements in the work area [19]. The third is to select a biological system that can rapidly reproduce the validity of observable effects [20,21]. Thus, observable physical changes in EMF measurements can be appreciated and adjusted in mathematical models that condition their intrinsic behavior [20]. A clear example is the Helmholtz coil system used for sensor validation, which generates a uniform magnetic field inside the coils and reduces the variability around their axes [22]. These systems are simple and convenient to validate different measurement systems, and with specialized programs, their behavior can be easily monitored [23].

Fermentation is a process in which microorganisms convert organic compounds into simpler compounds, such as alcohol or lactic acid, without oxygen [24]. The efficiency of fermentation processes is affected by various factors, such as temperature, pH, humidity, nutrient availability, and the presence of inhibitors or antimicrobial agents [24,25,26]. Recently, researchers have begun to investigate the potential impact of EMFs on fermentative processes. Although the role and influence of the magnetic field in this process are undeniable, there are many aspects to consider about its effects [12], which still need to be sufficiently understood [4,5,6,7,12,13,14,15,16,17,18].

On the other hand, one of the most functional and spontaneous biological systems that at the same time is reproducible is the fermentation of the cocoa bean. This process involves several microbial species [27] that act as a single ecosystem regulated by different environmental factors, such as humidity, pH, oxygen tension, and temperature [26]. These factors, in turn, are modulated by the metabolic activity of the microbial species that ferment the pulp [27]. The native microorganisms of the grain are naturally selected [28] and gradually colonize the pulp based on their metabolic potential [28,29]. Thus, the microbial metabolism produces notable external and internal physical changes in the bean [11].

Studies on the kinetic behavior and synergy of microbial populations during fermentation indicate the susceptibility of this process to environmental factors that often lead to poor fermentation, irreversibly reflected in the structural transformations of the cocoa bean [30]. These changes can be easily controlled using a simple temperature, humidity, pH, and Brix-monitoring device. Since variations in measurement procedures are one of the primary sources of bias in EMF studies, cocoa bean fermentation is an ideal study system to observe magnetic field effects on fermentation parameters. This research aims to validate an experimental prototype that generates low-frequency electromagnetic fields by monitoring CCN 51 cocoa bean fermentation.

To achieve the objective of the investigation, three essential stages were followed. The prototype and the digital signal connection system were designed in the initial step. Subsequently, in the second stage, the prototype underwent a rigorous validation of precision, reproducibility, and robustness against a gold standard for determining its ability to generate the expected electromagnetic emissions. Finally, the third stage consisted of evaluating the biological effects by monitoring key variables that characterize the fermentation process of CCN 51 cocoa beans. This comprehensive analysis examined the associations between the observed electromagnetic emissions and the identified variables.

This research study offers important information on the design, validation, and impact of electromagnetic emissions on cocoa beans. The results have substantial potential to improve the biotech and agricultural sectors by exploring novel approaches that improve cocoa production, contributing to the development of the chocolate industry.

## 2. Materials and Methods

### 2.1. Select Prototype Elements

In the selection of the aspects of the experimental prototype of electromagnetic emissions, Equation (1), the weighted factor method of the multi-criteria decision analysis (MCDA) was used [31]:(1)$$P_{i}=\overset{i}{\underset{k=n}{\sum }}W_{K}*S_{ki}$$
where P_i_ is the score of choice, W_k_ is the weighting of factor k, and Sk is the rating given to factor k in option n.

Five *k* factors (generation, electrical control, emission, and presentation of signals or data) were analyzed using expert criteria with three alternatives or components of each system or factor for selecting the elements according to their highest *P*. The number of loops in each coil, the magnetic field density, and the current that circulates through the coils were determined by Biot Savart law [32] (Equation (2)) to obtain field density readings of 0.5 to 110 mT.
(2)$$B=\frac{8\mu_{0}IN}{5\sqrt{5}\cdot a}$$
where I is the current in each coil, N is the number of turns in each, B is the density of the magnetic field generated by N loops on the x-axis, and a is the mean radius of the coil.

At the same time, the values of these variables corroborated by measuring instruments were obtained. In electrical measurements, a digital multimeter (UT56 LCD, UNI-T, Dongguan, China) was used, and ±(0.5%) of alternating current was used. The generated temperature was controlled with a digital thermometer (DeltaTrak 11050, Pleasanton, CA, USA) (±0.1 °C) considering normative guidelines for electrical safety in assembling mechanical, electrical, and electronic elements.

#### 2.1.1. Generation System and Electrical Support

In the design of the coils, materials such as plastic and wood were considered (*P* = 90%). Plastic was chosen for its insulating properties, while wood was selected for its mechanical support and stability. These materials were found to be suitable for the coil design due to their availability, cost-effectiveness, and compatibility with the desired performance of the system.

To power the electromagnetic emission system, an alternating current (AC) generator was chosen (*P* = 72%). The AC generator can deliver a sinusoidal waveform with a variable amplitude ranging from 0 to 100 V, operating at a frequency of 60 Hz. This specific waveform allowed the Helmholtz coils to generate a magnetic field with a density ranging from 0.55 to 110 mT, facilitating the desired experimental conditions for the study [33] (Figure 1).

The selection of the monoaxial Helmholtz matrix configuration, the careful design of the coils, and the choice of appropriate materials ensured a controlled and uniform magnetic field within the experimental setup. This configuration and the AC generator with the specified waveform parameters enabled the researchers to precisely investigate the effects of electromagnetic emissions on the cocoa beans’ fermentation process.

#### 2.1.2. Data Acquisition and Monitoring System

The selection of the high-precision Hall effect sensor (MLX92242, Melexis Co., Ltd., Tessenderlo, Belgium) (±100 mT) (*P* = 84.24%) was based on its ability to accurately measure the magnetic field strength, specifically the OMF (olfactory magnetic field). This sensor provides a wide measurement range and offers precise and reliable data for the electromagnetic emissions under investigation [34].

The Arduino nano board (Arduino LLC, China) (*P* = 92.02%) was chosen for signal acquisition. It functioned as the interface between the Hall effect sensor and the data processing system. The Arduino nano board was connected to the PC using an Ethernet cable, establishing a reliable and efficient data transfer and control connection [35].

The acquired data were visualized and analyzed, with the LabVIEW 2017 software (*P* = 90.23%) employed as a data presentation tool [36]. LabVIEW is a well-established software platform capable of acquiring, processing, and displaying scientific data. The software provides a user-friendly interface to observe and analyze the electromagnetic emission data captured by the Hall effect sensor and Arduino nano board. The data collected during the experiments were stored in a local file, ensuring data integrity and accessibility for further analysis [37,38] (Figure 1).

Low-cost, open-source boards such as the Arduino nano provide a cost-effective alternative combining advanced measurement and data acquisition capabilities. The Arduino nano offers high-quality experiments at a low cost, enabling researchers to conduct their studies without budget constraints [39]. The compatibility between the Arduino nano and LabVIEW 2017 software justifies their selection as hardware and software components. LabVIEW is a recognized software platform known for its reliability in scientific data analysis. It provides a comprehensive interface for acquiring, analyzing, and visualizing data, making it suitable for presenting the collected data and ensuring efficient data analysis.

### 2.2. Electrical Validation of the Prototype

#### 2.2.1. Linearity Measurements

The linearity of the measurements was evaluated by Student’s *t*-test (n-2 degrees of freedom, α = 0.05) and linear regression (least squares method) with a 95% confidence level [19,36] to corroborate the relationship between the input and output signal. The relationship between OMF oscillating magnetic field density OMF readings was determined using the MLX92242 mark Hall effect sensor (SEH) for the prototype and one-Tesla meter (PT2026, Ltd. GST, Japan) (accuracy 10 ppb) as the “Gold standard” [34] (Appendix A). Linearity was also evaluated with the hypothesis test for slope (b) and intercept (a) and the significance of the coefficients based on their standard errors (Ho: b = 0; Hi: b ≠ 0; Ho: a = 0; Hi: a ≠ 0) [36,40].

The linearity reveals adequate regression and proportionality in the measurements, and the linear model obtained is unaffected by systematic errors [36,38,40]. The response factors are similar, showing adequate precision of the measurements and the SEH reliability of the responses about the GEs (Table 1).

#### 2.2.2. Precision, Reproducibility, and Robustness Measurements

Precision was determined by OMF density measurements from the center of the Helmholtz coil to its edges in the workspace (p, α, z) according to the RS (response surface) design of experiments in the Design Expert v.13 programs (see Section 2.5 and Appendix A). The value of the measurements of OMF density was obtained by repeating this test under the same conditions by different analysts, days, and temperatures to different OMF densities (Appendix A). In each experiment, the mean, standard deviation, and variance of each response were obtained from the current variations in the coils (R1(5 mT), R2(42 mT), and R3(80 mT)) (Appendix A).

#### 2.2.3. Simulation and Validation of the Prototype

In the simulation, the magnetic field around the radial coordinates (ρ, α) starting from the center to the distant points of the axis (z) was calculated (Appendix A). From the polynomial equations of the models, three response surface graphs were obtained that describe the behavior of the field density in the work area [22]. The field density remains constant in this area and decreases as we move away from the central point (Appendix A). The COMSOL Multiphysics 6.0 program obtained this same behavior profile, where the exact coordinates were simultaneously simulated [41] (Figure 2).

The analysis of variance (ANOVA) conducted on the OMF density at different intensities (R1, R2, and R3) revealed statistically significant differences in the radial coordinates of the prototype, with a confidence level of 95.0%. The model F value for all responses was found to be significant, indicating that the model adequately explains the observed variations in the OMF density (Appendix A). Additionally, for the optimization, three random coordinates were selected and tested in the obtained polynomial equations, and simultaneously, the OMF density readings were obtained in the exact coordinates in RI, R2, and R3. Each OMF density measurement was replicated seven times at the selected coordinates (Appendix A). The average observations of each confirmation experiment (experimental coordinates) were within the prediction interval, which confirms the validity of the models obtained (Appendix A).

### 2.3. Experimental Processing

#### 2.3.1. Beans Processing

The cocoa pods of CCN51 cocoa beans obtained from local producers in the province of Santo Domingo, Ecuador, were disinfected in a 3% sodium hypochlorite solution for one hour. Subsequently, they were washed and manually opened under aseptic conditions to keep the grains clean and free of impurities or defects. Three kilograms of grains were placed in 5 L plastic fermentation boxes with ventral and lateral openings to facilitate the exudation of the beans [42] (Figure 3).

#### 2.3.2. Cocoa Fermentation Process

The fermentation process was carried out under controlled environmental temperature and humidity. The samples were placed in an orderly manner (according to the design described below) in the fermentation chamber and turned over periodically (every 24 h) to facilitate aeration [31,41]. During the periodic measurements of pH, temperature, and humidity, aseptic measures were always sought, and the measurement time was reduced to avoid interfering with the experiment results (Figure 3).

#### 2.3.3. OMF Expositions Process

Eight hours after fermentation, each experimental unit was placed in the center of the Helmholtz coils and subjected to different OMF densities for one hour according to the experimental design (0 mT, 5 mT, 42 mT, and 80 mT). The samples were placed vertically and close to the center of both rings, maintaining adequate separation between them. All treatments were properly monitored and randomized according to design (Figure 3 and Figure 4).

### 2.4. Monitoring Variables of the Process

#### 2.4.1. Continuous Variables

During the fermentation process, the CCN51 cocoa beans’ weight measurements were recorded with the digital balance brand SF-400 with a capacity of 10,000 g (±0.1 g), and the size of the beans was determined using the digital caliper (Vernier, Beaverton, OR, USA) (±0.01 mm). The pH, temperature, humidity, and Brix indices were determined in the mucilaginous pulp. The Brix concentration was measured using a digital refractometer (96801, HANNA Instruments, Woonsocket, RI, USA) (±0.2%); the pH temperature and humidity by using grain monitoring instrument kit (BlueLab Instruments, Tauranga, New Zealand) (±0.01 °C). For each experimental unit, three measurements (edges and center) were taken, with the average of each measure considered as the final value. All instruments were previously calibrated (Figure 4).

#### 2.4.2. Proportional Variables

At the end of the fermentation process, the cocoa beans were classified according to their degree of fermentation into well-fermented (Bw), moderately fermented (Bm), poorly fermented (Bp), and contaminated (Bc) using the cutting technique [41,42]. Bw was characterized by brown or reddish-brown cotyledonsss accompanied by well-open veins. Bm contained partially striated cotyledons with purple stripes on the edges; Bp comprised deep purple cotyledons, and Bc included unfermented black or gray cotyledons severely damaged by biological contamination [43,44,45]. This classification gave rise to the fermentation degree (Equation (3)).
(3)$$Fd=\frac{B\left(i\right)}{Bt}\times 100$$
where B represents beans’ average value; (i) is the classifications beans; Bt represents total beans.

Weight and size measurements were used to determine different proportional process variables such as rate grain weight loss (*W_l_*) (Equation (4)) and deformation rate (*D_r_*) (Equation (5)). The spectrometry determined the fermentation index through the absorbance ratio at 460 nm and 530 nm (Equation (6)) [45,46].
(4)$$W_{l}=\frac{Wo-Wf}{Wo}\times 100$$
(5)$$D_{r}=\frac{\frac{Xo}{Yo}-\frac{Xf}{Yf}}{\frac{Xo}{Yo}}\times 100$$
(6)$$F_{r}=\frac{Bx}{By}$$
where W represents the mean value of the weight of the beans (Equation (4)); Xo is the mean bean initial length value, Yo is the mean initial bean width value, Xf is the mean bean final length value, Yf is the mean bean final width value (Equation (5)), Bx represents the bean reading at 460 nm, and By represents the bean reading at 530 nm (Equation (6)).

#### 2.4.3. Microbial Analysis

Microbial viability was determined using ISO 6887 [47]. Ten grams of cocoa beans contained in 100 mL of sterile dilution of buffered peptone water (MERCK) were shaken vigorously for 10 min to obtain a uniform sample; the sample was filtered and serially diluted in the same water, then plated on selective media WL Nutrient agar to yeast; MRS agar to LAB; and glucose acetate agar (GAA) to AAB; and incubated for 48 h at 30 °C (all media came from Merck KGaA, Darmstadt, Germany) [25]. The microbial concentrations were calculated as the logarithm of the colony-forming units per gram (CFU g^−1^) by counting the number of colonies on the agar plates and multiplying it by the corresponding dilution factor according to the following Equation (Equation (7)) [48].
(7)$$N=log\left(\frac{\sum_{n=i}^{j}Cl}{Vi\times p\times d}\right)$$
where n represents replicas, i represents the subset of n; j represents a total of replicas, Cl represents the total number of colonies; V represents the inoculum volume, p represents the number of plates counted, and d represents minor dilution.

### 2.5. Experimental Design

A complete multifactorial design was carried out with a confidence level of 95% and, three repetitions for 48 experimental runs were designed in the Design Expert v.13.0.5.0 program. The factors were the OMF density (mT) and fermentation time (h). Four levels of OMF density were established (0, 5, 42, and 80 mT/60 min) as well as eight fermentation times in hours from (0, 24, 48, 72, 96, 120, 144, and 168 h). The effects were evaluated using the response variables of different growth parameters (Figure 3 and Figure 4).

The continuous variables, namely pH, temperature, humidity, and viability, were monitored during the seven days of fermentation and analyzed with ANOVA, and the proportions variables were analyzed by Poisson regression (Figure 3 and Figure 4).

## 3. Results and Discussion

### 3.1. Prototype Validation Results

#### 3.1.1. Electrical Validation

The spatial simulations of the electromagnetic field (EMF) carried out in this study, as shown in Figure 5a–c, yielded valuable information on the prototype’s performance. The results indicate that the magnetic flux density (B) readings at the center of the coils remained consistently stable across all tested scenarios (Figure 2 and Figure 5). This consistency is supported by the equations describing the density decline as a distance function, as shown in Appendix A. This confirms the remarkable stability, precision (Figure 5a,b), reproducibility (Figure 5c,d), and robustness (Figure 5e,f) of the prototype (Figure 1). The linearity of the magnetic field within the coils, as seen in Table 1, further supports these findings (Appendix A).

Maintaining a stable and predictable nature of the electromagnetic emissions generated by the prototype is crucial to guarantee its reliability and efficiency. This aspect is particularly significant when studying the effects of electromagnetic emissions on biotechnological systems such as cocoa beans. Previous studies emphasize the importance of stability and predictability in electromagnetic fields for precise and controlled experiments in various biotechnological applications [19,20,21,22,23].

The observed consistency in the magnetic field within the coils can be attributed to the design, precise location, and controlled current flow through the coils. Other researchers have also emphasized the importance of filtering the signal and using adequate material to maintain a uniform and stable magnetic field in Helmholtz coil configurations [21,22,23].

The uniformity of the magnetic field within the Helmholtz coils could be improved by implementing a polygonal coil design. Polygonal configurations strategically distribute coils, minimizing variations in magnetic field strength in the work area. This modification has the potential to improve the overall homogeneity of the field as well as the accuracy and robustness of the prototype [22,23].

It is advisable to recalculate the coil dimensions for a larger work area based on the desired radius. Increasing the coil dimensions proportionally to the desired working area ensures that the electromagnetic emissions cover a wider spatial range while maintaining the desired level of field uniformity. This adjustment allows the prototype to accommodate a larger footprint without compromising its optimal performance characteristics.

Numerous studies have shown that non-uniform magnetic fields can introduce variations in the response of biological systems, affecting the results of controlled biotechnological experiments [1,2,3,4,5,6,7,8,9,10,11,12,13,14,15,16,17,18]. By addressing non-uniformity through polygonal coil designs and dimension adjustments, the understanding and applicability of electromagnetic technologies in various biotechnology studies can be advanced [21,22,23]. These findings provide a solid foundation for the practical implementation of the prototype in studying the effects of electromagnetic emissions on cocoa beans. Thus, the stable and predictable nature of the electromagnetic emissions generated by the prototype makes it possible to assess their impact on the cocoa bean fermentation process accurately.

#### 3.1.2. Behavior of Process Parameters

The ANOVA models for the temperature, pH, humidity, and Brix responses demonstrate their significant nature, as indicated by their respective F-values of 94.26, 41.34, 38.69, and 74.56. The probability of obtaining such large F-values solely due to noise is exceptionally low at 0.01%. These findings confirm the robust significance of the models in explaining the observed variations in the responses. The *p*-values for the model terms A, B, and AB being less than 0.0500 in all four models indicate their significant contribution to the respective responses (see Appendix A).

During the fermentation of cocoa beans, the temperature and pH values of the beans were monitored and found to increase gradually, reaching their maximum values between 72 and 93 h (Figure 6b). Subsequently, the temperature and pH values decreased at the end of the fermentation period (168 h) (Figure 6a,b). Among the different OMF density treatments, the highest temperature values (51 °C) were observed at 5 mT and 42 mT, while the lowest temperature value (34 °C) was recorded at 80 mT (Figure 6a).

The temperature values during final fermentation were significantly different (*p* < 0.05) between the control and the OMF density treatments (0:43 °C, 5 mT:48 °C, 42 mT:48 °C, and 80 mT:36 °C) (Figure 6a). The maximum pH value (pH = 5) was recorded at 5 mT after 144 h, while the minimum pH value was observed at 80 mT after 48 h, with significant differences (*p* < 0.05) between the latter treatment and the rest of the treatments, including the control (Figure 6b). These results indicate that the OMF density treatments significantly impacted the temperature and pH levels during cocoa bean fermentation.

The moisture and Brix content gradually decreased, reaching minimum values at the end of the process, as depicted in Figure 6c,d. The highest Brix and moisture contents were recorded at 80 mT (Brix = 11Bx, Hr = 47%), whereas the lowest Brix and moisture contents were observed at 5 mT and 42 mT (Brix = 4Bx, Hr = 42%), with significant differences when compared to the control (Brix = 8, Hr = 45%) (Figure 6c,d).

The OMF density treatments significantly impacted the temperature, pH, moisture, and Brix content of cocoa beans during fermentation. The highest temperature values were observed at 5 mT and 42 mT, while the lowest was recorded at 80 mT. The highest Brix and moisture content was observed at 80 mT, and the lowest was recorded at 5 mT and 42 mT. These results indicate that the OMF density treatments significantly affect these variables during the fermentation of cocoa beans.

The optimum pH range for cocoa bean fermentation was found to be between 4.5 and 5.5, which favored the growth of lactic acid bacteria and the production of desirable flavor compounds [30]. In conclusion, the pH behavior during cocoa bean fermentation is a complex process that varies based on the cocoa variety, fermentation stage, and microbial activity. Additionally, it can be influenced by the pre-conditioning of the pulp, biochemical constituents, and polyphenolic constituents during fermentation [26,30,47,48].

Various scientific articles have discussed the alterations in pH that occur during cocoa fermentation [26,47,48]. In one such study, Mulono et al. [49], reported that the pH of cocoa beans increased from an initial pH of 4.4 to a maximum of 6.4 after 72 h of fermentation. However, after 96 h of fermentation, the pH decreased due to the production of organic acids by lactic acid bacteria. The authors noted that the pH of cocoa beans could vary depending on the microbial community structure and the specific strains of bacteria involved in fermentation.

Afoakwa et al. (2013) [26] also reported a rise in cocoa bean pH during the initial stages of fermentation due to the breakdown of glucose and fructose. However, after 48–72 h, pH levels dropped owing to the production of organic acids such as acetic and lactic acid by lactic acid bacteria. In a subsequent study in 2014, the same authors measured pH changes during cocoa fermentation and observed a decrease in pH levels after the first and second days of fermentation, with pH values declining from 6 to 3 and 4 at the end of the process due to the production of organic acids by lactic acid bacteria [50], which was in line with other studies [30,48]. Several authors emphasize the impact of the structure of microbial communities and specific bacterial strains on the pH of the cocoa bean, indicating that if the pH becomes too acidic too soon (pH < 4.5), there will be a final reduction in flavor precursors and a final product that is too acidic [25,27].

Herrera-Rocha et al. [51] conducted a study to examine the influence of temperature on the quality of cocoa bean fermentation and its impact on various selected physicochemical characteristics. They determined that a temperature range of 45–50 °C was optimal for cocoa bean fermentation, resulting in desirable flavor compounds and reduced levels of undesirable flavor compounds. Cortez et al. [52] found that cocoa fermentation leads to temperatures exceeding 45 °C. This increase in temperature triggers the activation of native enzymes and the denaturation of proteins, resulting in a notable impact on the development of flavor during cocoa fermentation. Furthermore, this research revealed that the increase in temperature is closely related to the formation of the key aroma molecules responsible for improving the sensory profile and overall quality of cocoa beans.

In cocoa fermentation, Brix degrees can be used to monitor the conversion of sugars into organic acids and alcohol by microorganisms during the fermentation process [49,50]. Velásquez-Reyes et al. [53] studied differences between different cocoa bean varieties in the profile of volatile and non-volatile compounds in the process from fermentation to liquor. The authors found that Brix levels decreased during the fermentation due the metabolic activity of microorganisms involved in the process. Hernandez et al. [54] studied the physicochemical and microbiological dynamics of the fermentation of CCN51 cocoa material in three stages of maturity and demonstrated that microorganisms present in certain states were more effective in metabolizing sugars compared to those in other states. In the same way, it is evident that a gradual decrease in the degrees with the advance of fermentation is due to the consumption of sugars by microorganisms.

In their research, Camu et al. [25] explored the effects of both fermentation and drying processes on the microbial communities associated with cocoa beans. They found that Brix levels increased during fermentation and peaked around days 3–4 before decreasing during drying, likely due to the loss of water and sugars in the beans and the consumption of sugars by microorganisms during fermentation.

Overall, these studies highlight the critical role of temperature, pH, Brix, and humidity in microbial growth during cocoa bean fermentation. The optimal levels of these parameters may vary depending on the specific microbial strains and environmental conditions during fermentation. Thus, the results suggest that EMFs could indirectly affect cocoa bean physicochemical parameters.

#### 3.1.3. Bean Weight and Fermentation Rate Behavior

ANOVA results for the Wi, Fi, and Di responses demonstrate the significance of the models. The F-values for Wi, Fi, and Di are 77.37, 31.60, and 9.59, respectively. These high F-values indicate the models’ strong significance, with only a 0.01% probability that the observed F-values are due to noise alone. The A, B, and AB terms have a significant effect on the respective response variables (see Appendix A).

The weight loss of cocoa beans during fermentation increased over time and stabilized between 96 and 168 h, with the highest rates observed as shown in Figure 7a. The 80 mT treatment showed the lowest Wi rates (Wi = 9%), with significant differences between the control (Wi = 16%) and the rest of the treatments (5 mT: Wi = 16%, 42 mT: Wi = 14%) (Figure 7a).

Cocoa beans’ Fi increased until stable maximum values were reached between 120 and 168 h (Figure 7b). At the final fermentations, the highest Fi value was observed at 5 mT (Fi = 2.5%), with significant differences compared to the control and the rest of the treatments (42 mT: Fi = 2%, 80 mT: Fi = 1.8%). Conversely, the lowest Fi values were obtained at 80 mT (Fi = 1.5%) at 144 h, with the same statistically significant differences (*p* < 0.05) (Figure 7b).

The deformation of cocoa beans increased over time and exhibited the highest standard deviation compared to other variables (Figure 7c). In the final stages of fermentation, the 42 mT treatment had the highest Di rates (Di = 4.83%), whereas the 80 mT treatment had the lowest (Di = 2.03%), with no significant difference (*p* > 0.05) observed between the control and the other treatments. At the end of fermentation, the values for the treatments and controls were nearly identical (Figure 7c).

The highest Bw rates (dependent on coloration) were observed at 5 mT (Fd = 94%), with the lowest rates recorded at 80 mT (Fd = 53%). These differences were significant (*p* < 0.05) compared to the control (Fd = 80%), as shown in Figure 7d. Furthermore, the highest rates of Bp were observed at 80 mT (Fd = 46%), with significant differences between the control (Fd = 10.5%) and the other treatments (5 mT: Fd = 4%, 45 mT: Fd = 7%) (Figure 7d). Additionally, the control treatment exhibited significantly (*p* < 0.05) higher rates of contaminated grains (Fd = 12%) compared to the other treatments (5 mT: Fd = 1%, 45 mT: Fd = 1%, 80 mT: Fd = 0%) at the end of fermentation (Figure 7d).

These results indicate that the OMF density treatments significantly affect weight loss (Wi) and fermentation index (Fi) as well as fermentation degree (Fd). However, there was no significant difference in the deformation index between the control and other treatments.

Camu et al. [25] documented that the weight loss experienced by cocoa beans during fermentation predominantly originates from the degradation of carbohydrates, proteins, and lipids. Weight loss increases rapidly during the first few days of fermentation, with a typical behavior of 5% to 15%, and then gradually levels off towards the end of the process. Another study by Hernandez-Hernandez et al. [55] reported that weight loss during cocoa fermentation is due to the loss of water and the breakdown of organic matter in the bean. These investigations corroborate the notion that weight loss is an important factor that affects the quality of cocoa beans during fermentation. Still, the exact rate of weight loss and duration of fermentation can depend on several factors, such as the type of cocoa beans, the fermentation method used, and environmental conditions [24,27,28,29,30]. Changes in the fermentation rate, color, texture, and flavor of the beans also accompany Wi [19,20,21,22,23,24,25,26,27,28,29,30].

Camu et al. [27] reported that the fermentation rate of cocoa beans varied depending on the type of microbial community present during fermentation. The authors found that the fermentation rate was higher in the presence of lactic acid bacteria than in the presence of yeasts or acetic acid bacteria. They also noted that the rate of temperature increase was slower in fermentations dominated by lactic acid bacteria, suggesting that these bacteria have a more moderate effect on the fermentation process. This finding is consistent with the idea that the microbial community plays an important role in the fermentation process of cocoa beans. Thus, the presence of lactic acid bacteria can affect the fermentation rate and the temperature increase during fermentation, which can ultimately affect the quality of the cocoa beans.

According to several authors [28,29,30], during the fermentation process of cocoa beans, the beans undergo deformations due to the rupture of the beans’ cell walls and the release of water and sugars from the pulp. The gradual softening of the pulp and the reduction in the size of the bean is attributed to the activity of microorganisms and enzymes, which contribute to the production of flavor precursors and the development of the characteristic flavor of cocoa. Yeasts, acetic acid bacteria, and lactic acid bacteria stimulate enzymatic activity to hydrolyze sugars, organic acids, proteins, and polyphenols in cocoa beans, causing physical and chemical changes in the beans. These changes lead to the deformation of the beans during fermentation [54,55,56,57].

In the study conducted by Guéhi et al. [56], it was noted that cocoa subjected to a 4-day fermentation period exhibited a higher percentage of purple beans, reaching approximately 45%. Additionally, the presence of moldy beans was observed to be around 1%, while approximately 10% of the beans showed signs of discoloration. The formation of brown beans increased from 16% to 50% depending on the fermentation duration and process. Several authors highlight the main changes that determine Fd and the final quality of cocoa that occurs during fermentation given the great number of factors that affect the process including microbial populations [24,25,26,27,28,29,30]. Other researchers also found that using a starter culture of LAB resulted in higher fermentation indices and darker bean colors than spontaneous fermentation, indicating that LAB may play an essential role in the fermentation process [24,25].

According to another study by Afoakwa et al. and Saputro et al. [26,55], the fermentation rate of cocoa beans is highest during the first 48 to 72 h. During this time, the temperature of the cocoa mass increases due to microbial activity, suggesting that the initial stages are characterized by a more intense fermentation process, which promotes the development of chocolate flavor. De Vuyst and Weckx as well as Guzmán-Alvarez et al. [28,56] highlighted the role of organic acids, specifically acetic acid and lactic acid, which are produced by microorganisms in the cocoa pulp, in the deformation of cocoa beans during fermentation. The researchers observed that this increase led to a decrease in the size of the beans and a color change, which were indicators of the degree of fermentation and the quality of the beans.

The behavior of Wi, Fi, Di, and Fi is also influenced by microbial activity; consequently, the effect of EMF in variation in these rates may also be conditioned by changes in the growth of microbial groups.

#### 3.1.4. Microbial Group Behavior

The ANOVA models for LAB, AAB, and Y responses exhibit significance, as indicated by the respective F-values and the low probability of obtaining such values due to noise alone. The *p*-values for the model terms A, B, and AB being less than 0.0500 confirms their significance in all three models. These findings strengthen the reliability and validity of the models, indicating that they provide significant results for understanding the relationship between the factors and the observed responses (see Appendix A).

In the control treatment, the concentration of lactic acid bacteria (LABs) reached its maximum point between 24 and 48 h, with values of 5.72 and 5.92 CFU/g, respectively, and decreased to a minimum value of 1.52 CFU/g towards the end of fermentation, corresponding to a reduction of 4.4 log cycle (Figure 8a). In the rest of the treatments, much lower LAB concentration values were obtained, with significant differences compared to the control (*p* < 0.05) (Figure 8a). The lowest LAB values were observed in the 80 mT treatment (1.05 CFU/g), while the highest values were observed in the 5 mT treatment (4.23 CFU/g) at 72 h, with significant differences between treatments (*p* < 0.05) (Figure 8a).

The maximum point of acetic acid bacteria (AAB) was observed in the 5 mT treatment between 96 (7.75 CFU/g) and 120 h (7.96 CFU/g), with significant differences compared to the control (5.69 CFU/g, 5.49 CFU/g) at the same time (Figure 8b). The lowest AAB values were observed in the 80 mT treatment at the final stages of fermentation (2.10 CFU/g), with significant differences compared to the control (4.33 CFU/g) and other treatments (5 mT: 4.35 CFU/g, 42 mT: 5.7 CFU/g) (Figure 8b). The most significant reduction in acetic acid concentration was observed in the 5 mT treatment, with a reduction of 3.4 log cycles (Figure 8b).

During the first 48 h of fermentation, the 5 and 42 mT treatments showed the highest Ys growth rates (7 CFU/g), with significant differences (*p* < 0.05) compared to the control and other treatments (Figure 8c). By the end of fermentation, the concentration of yeasts had reduced to 2.5 CFU log cycle, with the most significant reduction observed in the 80 mT treatment at 4.2 CFU log cycle reduction (Figure 8c).

The microbial population dynamics during cocoa bean fermentation are highly complex and involve the sequential activation of several microbial populations. Lasting for the first 24 to 36 h, yeasts are the dominant microorganisms, rapidly consuming the sugars in the pulp surrounding the cocoa beans and producing ethanol and carbon dioxide as byproducts. The ethanol produced by yeasts creates an acidic environment that favors the growth of LAB. LAB then dominates the fermentation process and produces lactic acid, which further lowers the pH and inhibits the growth of yeasts. As the fermentation progresses, AAB becomes more active and oxidizes the ethanol produced by yeasts to acetic acid. The acetic acid produced by AAB contributes to the development of the characteristic flavor and aroma of chocolate. However sequential activation of these microbial populations is influenced by various factors, such as the type of cocoa bean variety, the fermentation conditions, and the presence of other microorganisms [24,25,26,27,28].

The results suggest that the application of magnetic fields during cocoa fermentation affects the growth and population dynamics of yeast, lactic acid bacteria, and acetic acid bacteria. The 5 mT and 42 mT treatments showed the highest yeast growth rates during the first 48 h.

Bubanja et al. [58] investigated the influence of low-frequency magnetic field regions on the respiration and growth of *Saccharomyces cerevisiae.* The study found statistically significant differences in cumulative oxygen consumption, cumulative carbon dioxide production, and *S. cerevisiae* cell number, which was attributed to the MF-induced stimulation of microbial growth and activity. De Andrade et al. [5] conducted a study to investigate the effect of an extremely low-frequency (ELF) magnetic field on bioethanol productivity by *S. cerevisiae* in an unconventional bioreactor. The study found a 33% increase in bioethanol production, which was consistent with a stimulatory effect of the magnetic fields on plasma membrane H+-ATPase activity. The observed effects were attributed to the EMF-induced enhancement of enzyme activity and regulation of gene expression. Similar studies have shown that applying magnetic fields can positively impact microbial communities by increasing microbial growth, metabolic activity, diversity, and function, improving the efficiency of biotechnological processes such as anaerobic digestion, wastewater treatment, and constructed wetlands [4,5,6,7,18].

Zieliński et al. [59] reported increased the microbial diversity and degradation efficiency of organic matter in anaerobic digestion bacteria exposed to a magnetic field. Zaidi et al. [60] found that applying a magnetic field to wastewater treatment systems led to the removal of pollutants, while Ma et al. [61] reported that an external static magnetic field at 14 mT enhances the reduction of antibiotic-resistance genes during pig manure composting.

Hu et al. [62] investigated the impact of a static magnetic field on electron transport and microbial community shifts in the nitritation sequencing batch process. The study found that a static magnetic field can accelerate the start-up of the nitritation process and improve its performance by changing the microbial community structure and improving the HAO activity, the Cyt c content, and ETSA as well as the energy generation of microorganisms. Lyu et al. [63] reported that a magnetic field at 10 mT could increase the abundance and diversity of microbial communities, promoting the secretion of tryptophan and aromatic proteins.

On the other hand, Rakoczy et al. [64] conducted a study to investigate the effect of a ferrofluid and rotating magnetic field (RMF) on the growth rate and metabolic activity of a wine yeast strain. The study found that exposure to RMF resulted in a decrease in yeast cell numbers and inhibited their metabolic activity. Bayraktar [65] also found that magnetic field treatment at 5 mT for 30 min inhibited the reproduction and enzymatic activity of *S. cerevisiae* [21]. Similarly, in this study, the EMF effect significantly reduced the LAB population in all treatments. Meanwhile, at 80 mT, the lowest growth rates of all the microbial groups compared were produced.

Other studies have demonstrated the ability of magnetic fields to enhance cocoa fermentation. Sudarti et al. [66] reported that exposure to magnetic fields significantly improved cocoa fermentation compared to the control group, as reflected in the reduction of pH and the increase in fermentable sugar content. Similarly, Guzman et al. [10] found that EMF improved the yield and quality of fermented cocoa beans. While there are differences in the methodology used in both studies compared to the other studies reviewed, the conclusive results indicate that electromagnetic fields have a profound impact on microbial metabolism.

Therefore, exposure to magnetic fields could alter the microbial dynamics in cocoa fermentation, which is reflected in a reduction of pH, an increase in the content of fermentable sugars, higher production of acetic acid and ethanol, and possibly better organoleptic characteristics of chocolate. The observed differences in microbial population dynamics suggest that magnetic fields can potentially be used as a tool to manipulate the fermentation process and improve the quality of cocoa products. Moreover, it is noteworthy that the optimal intensity, duration, and other parameters of the magnetic field may vary depending on the specific fermentation process and the microbe being used [58,59,60,67].

However, further studies are required to determine the optimal exposure parameters of magnetic fields and to evaluate the long-term effects on the quality of chocolate produced from cocoa beans fermented with magnetic fields. In this sense, the findings of this research suggest that under the conditions of this study, electromagnetic fields influenced microbial populations by inhibiting the growth of LAB and stimulating AAB and yeast, thereby causing a rebound effect in the analyzed process variables.

## 4. Conclusions

This study successfully developed a prototype to investigate the effects of electromagnetic emissions on bioprocesses, specifically the fermentation of cocoa beans. Magnetic flux density readings remained consistently stable at the center of the coils, demonstrating excellent linearity, reproducibility, and accuracy. These findings provided reliable evidence of the impact of electromagnetic fields on various parameters of the cocoa bean fermentation process.

The observed effects of the electromagnetic field on cocoa bean fermentation can be attributed to EMF-induced growth stimulation, microbial activity, and enzyme activity. However, more research is needed to fully understand the underlying mechanisms responsible for these effects on cocoa bean microbial populations and their implications for the chocolate industry.

It is widely recognized that the sequential activation of different microbial populations is a critical feature of the fermentation process. Understanding these dynamics is essential to produce high-quality chocolate. Therefore, continuous research on the effects of electromagnetic fields on cocoa bean fermentation will contribute to advancing the knowledge and optimization of chocolate-production processes.

## Figures and Tables

**Figure 1 foods-12-02539-f001:**
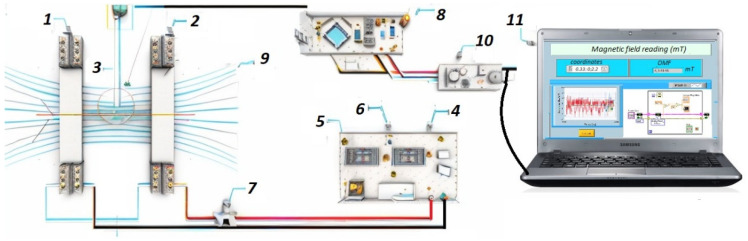
MEF system. (1) Coil 1 (2); coil 2; (3) magnetic sensor (Hall effect Tesla meter); (4) variable voltage source; (5) voltage indicator; (6) current indicator; (7) ammeter; (8) Arduino nano; (9) magnetic field flux lines; (10) sensor module; (11) LabVIEW data output signal on the PC.

**Figure 2 foods-12-02539-f002:**
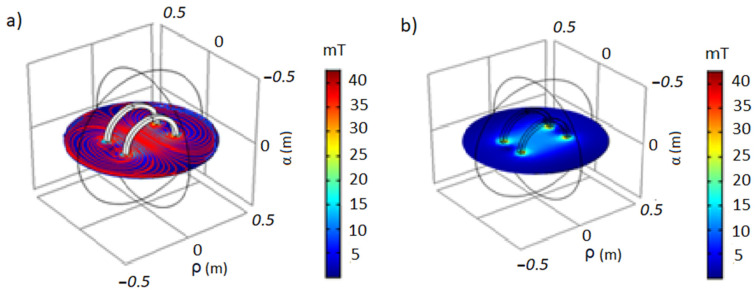
Flux density vs. distance spectra cross-section simulation in COMSOL Multiphysics (radial axis: ρ, α; axial axis: z): (**a**) 5 mT and (**b**) 42 mT.

**Figure 3 foods-12-02539-f003:**
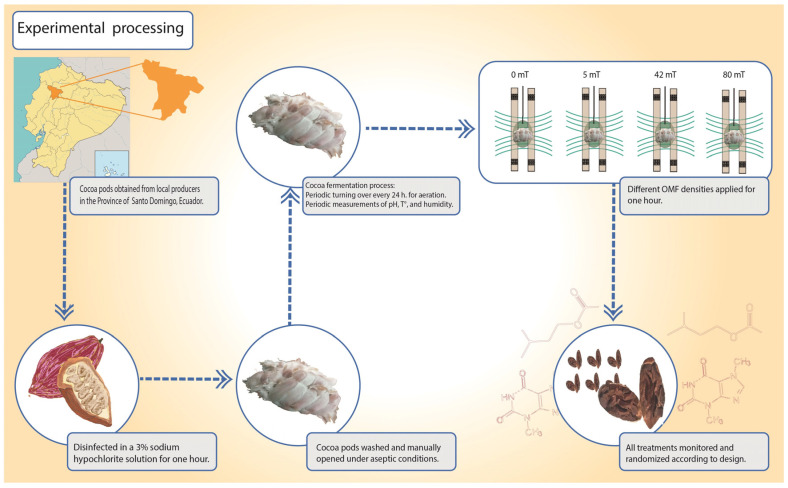
Schematic drawing of the experimental procedure for cocoa beans.

**Figure 4 foods-12-02539-f004:**
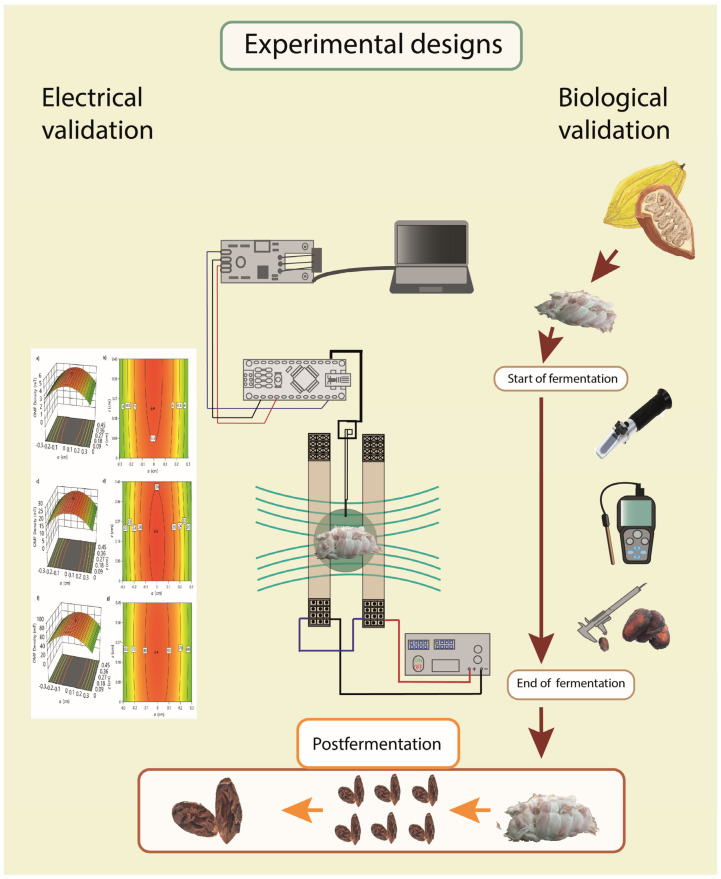
Schematic drawing of experimental designs. Electrical and biological validations in Design Expert software v13.

**Figure 5 foods-12-02539-f005:**
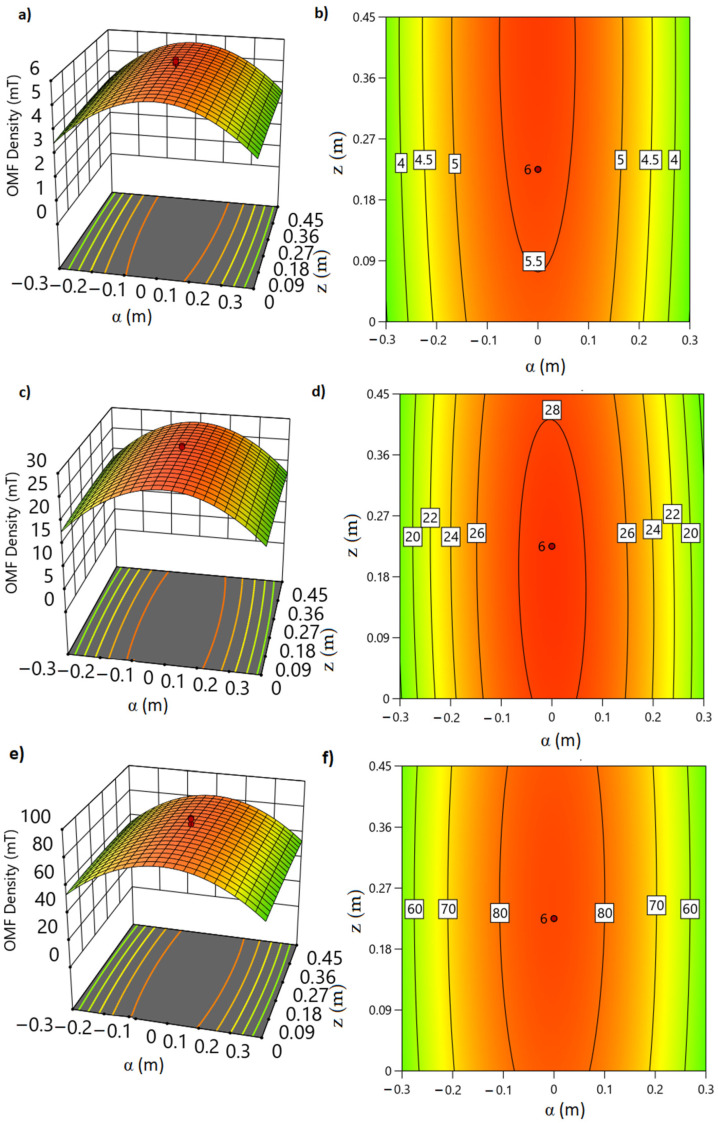
Response surface (RS) and contour graphs (CG). Modeling of the OMF density in the workspace as a function of days (**a**,**b**), analysis (**c**,**d**), and temperature (**e**,**f**). RS: vertical axes, OMF density; horizontal axes, α and z; CG: vertical axes, α; horizontal axes, z; contour line, OMF density. ρ maintained in 0 values.

**Figure 6 foods-12-02539-f006:**
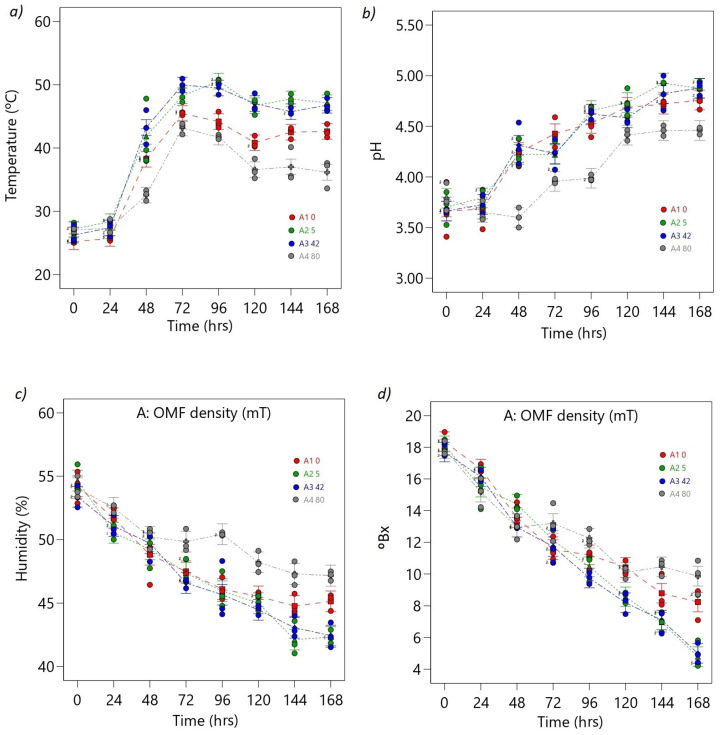
ANOVA interaction plots. Continuous responses: (**a**) temperature, (**b**) pH, (**c**) humidity, and (**d**) Brix. The ordinal factor, time, is displayed continuously, and the nominal OMF density factor (A) is presented discreetly as separate lines (A1: 0 mT red line, A2: 5 mT green line, A3: 40 mT blue line, A4: 80 mT gray line). The dots represent the values that the response variable takes around the mean; the midpoint of the bar represents the mean value; overlapping bar lines between levels indicate *p* > 0.05; no overlap of bar lines between levels indicates *p* < 0.05.

**Figure 7 foods-12-02539-f007:**
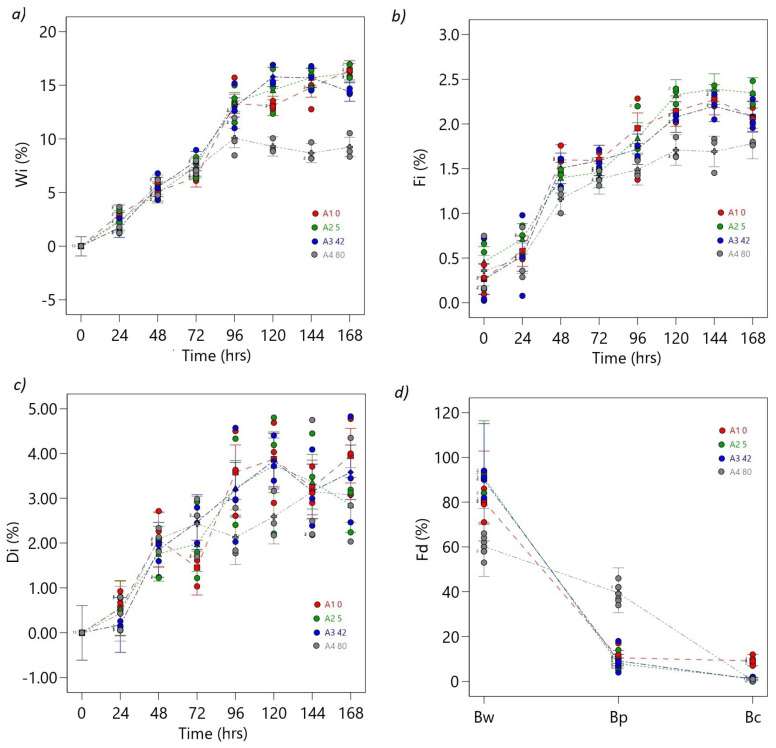
ANOVA interaction plots. Proportional responses: (**a**) Wi, (**b**) Fi, (**c**) Di and (**d**) Fd. The ordinal factor, time, is displayed continuously, and the nominal OMF density factor (A) is presented discreetly as separate lines (A1: 0 mT red line, A2: 5 mT green line, A3: 40 mT blue line, A4: 80 mT gray line). The dots represent the values that the response variable takes around the mean; the midpoint of the bar represents the mean value; overlapping bar lines between levels indicate *p* > 0.05; no overlap of bar lines between levels indicates *p* < 0.05.

**Figure 8 foods-12-02539-f008:**
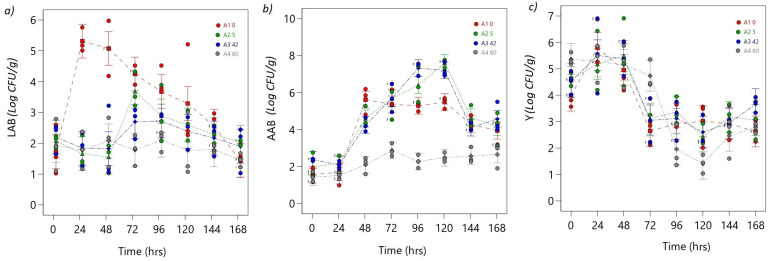
ANOVA interaction plots. Continuous responses: (**a**) LAB, (**b**) AAB, and (**c**) Y. The ordinal factor, time, is displayed continuously, and the nominal OMF density factor (A) is presented discretely as separate lines (A1: 0 mT red line, A2: 5 mT green line, A3: 40 mT blue line, A4: 80 mT gray line). The dots represent the values that the response variable takes around the mean; the midpoint of the bar represents the mean value; overlapping bar lines between levels indicate *p* > 0.05; no overlap of bar lines between levels indicates *p* < 0.05.

**Table 1 foods-12-02539-t001:** ANOVA for field densities and regression coefficients of the linear model; *t*-test, for the slope of the model, (slope: b = 0.904, null hypothesis: b = 0, alternative hypothesis: b ≠ 0); the probability of being b = 0 is low (*p*-value = 0.000); for the intercept (a = 1.641), the value with a low probability of being is a = 0 (*p*-value = 0.000), (slope ≠ 0; intercept ≠ 0). The value of the intraclass correlation coefficient is (ICC = 0.994).

Fountain	Sum of Squares	Gl	Middle Square	F-Reason	*p*-Value
Model	50,596.5	1	50,596.5	25,844.27	0.0000
Residue	164.451	84	1.95775		
Total (Corr.)	58,444.5	84			
	Least Squares	Standard	Statistical		
Parameter	Dear	Error	T	*p*-value	
Intercept (a)	1.641	0.287	5.708	0.0000	
Slope (b)	0.904	0.005	160.762	0.0000	
Correlation Coefficient (r)	R-Square	Standard error of the set.	Average absolute error	CCI	
0.998	0.996	1.39	1.13	0.994	

## Data Availability

Data is contained within the article or Appendix A.

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
