# Peer review of "Experimental Prototype of Electromagnetic Emissions for Biotechnological Research: Monitoring Cocoa Bean Fermentation Parameters"

_foods, 2023, doi:10.3390/foods12132539_

Round 1
Reviewer 1 Report
The authors investigated the effects of electromagnetic fields on the fermentation of cocoa beans. The topic is interesting, but the manuscript is not well written and does not meet the quality. Therefore, the manuscript is not worthy of peer review.
There are too many paragraph breaks and too many typos (e.g., ± (0.5%)). Also, figure numbers cited in the text do not match the corresponding figure numbers. Figure captions do not match figures.
Reviewer 2 Report
The authors investigated a prototype of electromagnetic emissions for bioprocesses studies and application in cocoa beans fermentation. The authors failed to used line numbering making it difficult to refer to the text. The authors should correct errors in language and punctuation.
Comments
Abstract
Correct the first sentence.
EMF: electromagnetic fields? Define EMF
Section 2.4.1
How was the weight taken? The authors used 3 kg of grains for fermentation, and the digital balance used has a capacity of 1000g (±0.01g).
Result
Section 3.1, page 9, 3rd paragraph, correct the last sentence. The authors should correct figure numbering
Section 3.2, page, Figure 7 and 8 are not in the manuscript
Page 11, 1st paragraph, correct the 1st sentence
Correct citation style
The manuscript need moderate editing of English language
Reviewer 3 Report
The authors evaluated the Experimental prototype of electromagnetic emissions for biotechnological research: Monitoring cocoa beans fermentation parameters.
Some minor remarks follow.
In the abstract.... EMF-exposure... EMF what is this acronym?
from a double Perspective...why capital P?
Guehi et al. (2010) [54], observed that Cocoa fermented....why capital C?
According to several authors [28 – 30], During....why capital D?
O2 consumption, cumulative CO2...Please write in the correct form
and S. cerevisiae cell number...Please write the pathogen in italics
and growth of Saccharomyces cerevisiae...Please write the pathogen in italics
activity of Saccharomyces cerevisiae...Only the first time in the text we write the whole name of pathgen
Please write all the references with the same way and according journal's guidelines.
Reviewer 4 Report
The topic of the manuscript is interesting given the attention that data acquisition and monitoring systems are receiving from Academy and industry for application in food production scenarios using modern technologies. In addition, this topic fits the scope of the Journal. After a careful revision, the following comments are provided for the enhancement of the manuscript.
A keyword to include is “monitoring” for a higher visibility of the paper in this field.
A common practice in scientific papers consists on placing a paragraph at the end of the introductory section describing in a brief manner the structure of the rest of the manuscript. This contributes to the readability of the document and is suggested to be included in the present case.
Regarding the data acquisition and monitoring system, combining low-cost Arduino boards with well-known LabVIEW software is a positive feature. Nonetheless, some questions arise. To begin with, the development of this tailored equipment instead of using commercial systems should be briefly commented and justified. For example, applying open-source and low-cost boards like Arduino is a growing trend nowadays and could be supported by recent publications:
- Integration of open source hardware Arduino platform in automation systems applied to Smart Grids/Micro-Grids. Sustainable Energy Technologies and Assessments 2019, 36, 100557. DOI: 10.1016/j.seta.2019.100557
- Identification and Classification of the Tea Samples by Using Sensory Mechanism and Arduino UNO. Inventions 2021, 6, 94. https://doi.org/10.3390/inventions6040094
- An Intelligent IoT-Based Food Quality Monitoring Approach Using Low-Cost Sensors. Symmetry 2019, 11, 374. https://doi.org/10.3390/sym11030374
- Monitoring Mung Bean’s Growth using Arduino. Procedia Computer Science 2021, https://doi.org/10.1016/j.procs.2021.01.016
Moreover, additional information about the designed and validated system should be given. For instance, why have the authors choose the Nano board instead of other models? Is there some boundary about the size of the system? How is Arduino connected to the PC (USB, Ethernet, WiFi)? Another aspect is how data is stored (local file, database, cloud). On the other hand, the version of LabVIEW is 2017, which could be seen as a bit old nowadays. To sum up, the authors should enhance the description of this system for a proper presentation of the materials and methods. A screenshot of the LabVIEW interface would also enrich this section.
In Figure 1 there is a small mistake in the title, namely, “emissios” should be replaced by “emissions”.
Some figure captions lack the terminal period (punctuation).
In subsection 2.3, a set of steps for beans processing and further stages are described sequentially. This is correct, but adding a flowchart to illustrate the described steps would boost the visual features of the paper as well as the understanding of this relevant part of the manuscript.
In the third section, indicating the publication year of cited papers is not required. For example, “Mulono et al. (2016) [47]” should be replaced by “Mulono et al. [47]”.
The Results or, alternatively, the Conclusion section, should comment, at least in a brief manner, about the developed monitoring system, not only about the effect of electromagnetic fields on cocoa processes. In fact, a relevant part of the paper is devoted to describe and validate such monitoring system, which is a valuable contribution.
The section Data Availability Statement should be revised in order to remove the text that the template of the Journal provides.
The format of references must be slightly revised to match the template.
Round 2
Reviewer 4 Report
The new version of the manuscript has taken into account the reviewers comments in a proper manner.